# Identification and Characterization of 5-HT Receptor 1 from *Scylla paramamosain*: The Essential Roles of 5-HT and Its Receptor Gene during Aggressive Behavior in Crab Species

**DOI:** 10.3390/ijms24044211

**Published:** 2023-02-20

**Authors:** Xinlian Huang, Yuanyuan Fu, Wei Zhai, Xiaopeng Wang, Yueyue Zhou, Lei Liu, Chunlin Wang

**Affiliations:** 1School of Marine Science, Ningbo University, Ningbo 315832, China; 2Department of Marine Medicines and Biological Products, Ningbo Institute of Oceanography, Ningbo 315832, China

**Keywords:** *Scylla paramamosain*, biogenic amines, 5-hydroxytryptamine, *Sp5-HTR1*, aggressive behavior

## Abstract

Biogenic amines (BAs) play an important role in the aggressive behavior of crustaceans. In mammals and birds, 5-HT and its receptor genes (5-HTRs) are characterized as essential regulators involved in neural signaling pathways during aggressive behavior. However, only one 5-HTR transcript has been reported in crabs. In this study, the full-length cDNA of the 5-HTR1 gene, named *Sp5-HTR1*, was first isolated from the muscle of the mud crab *Scylla paramamosain* using the reverse-transcription polymerase chain reaction (RT-PCR) and rapid-amplification of cDNA ends (RACE) methods. The transcript encoded a peptide of 587 amino acid residues with a molecular mass of 63.36 kDa. Western blot results indicate that the 5-HTR1 protein was expressed at the highest level in the thoracic ganglion. Furthermore, the results of quantitative real-time PCR show that the expression levels of *Sp5-HTR1* in the ganglion at 0.5, 1, 2, and 4 h after 5-HT injection were significantly upregulated compared with the control group (*p* < 0.05). Meanwhile, the behavioral changes in 5-HT-injected crabs were analyzed with EthoVision. After 0.5 h of injection, the speed and movement distance of the crab, the duration of aggressive behavior, and the intensity of aggressiveness in the low-5-HT-concentration injection group were significantly higher than those in the saline-injection and control groups (*p* < 0.05). In this study, we found that the *Sp5-HTR1* gene plays a role in the regulation of aggressive behavior by BAs, including 5-HT in the mud crab. The results provide reference data for the analysis of the genetic mechanism of aggressive behaviors in crabs.

## 1. Introduction

“Aggressive behavior” refers to the behavior of fighting between individuals of the same species to determine the dominant and subordinate positions when they meet, a typical social behavior. The winner of the aggressive encounter gains more and longer-term access to ecological resources and mating options, while the loser misses the opportunity to share various resources equitably [1,2]. Crustaceans are particularly aggressive in intraspecies competition due to their armored chelipeds or tentacles. The occurrence of competition in culture and breeding can seriously affect their survival rate and production, resulting in economic losses [3]. Previous studies showed that the aggressive behavior of crustaceans is affected by environmental factors such as temperature, dissolved oxygen, and light [4,5,6], and nonenvironmental factors including size, sex, density, prey resources, and prefighting experience [7,8,9,10].

In addition, the aggressive behavior of crustaceans is regulated by neurotransmitters [11]. Biogenic amines (BAs) are widely distributed in the central nervous system and peripheral organs of crustaceans, and act as neurotransmitters or hormones to transmit various messages. The neurotransmitters secreted by the neuroendocrine system play important roles in aggressive behavior, and the system is a key decision-making center for appropriate responses during the aggressive interactions of crustaceans [12].

Neurotransmitter 5-HT, also known as serotonin, is widely distributed in the central and peripheral nervous tissues of crustaceans [13], and is an inhibitory neurotransmitter. Serotonin (5-HT) plays a modulatory role in numerous physiological processes such as food intake, sleep, and aggressiveness [14]. Specifically, 5-HT is the most dominant neurotransmitter affecting agonistic behavior [15], while other neurotransmitters may first modulate agonistic behavior by affecting 5-HT levels in the organism [16]. Injections of 5-HT modulated crayfish muscle tone and rendered them more aggressive than injections of other BAs did [17,18].

Furthermore, the presence of specific receptors on the presynaptic and postsynaptic membranes is essential for 5-HT to produce any physiological effects [19]. On the basis of the molecular structure, mechanism of action, and different functions, the 5-HT receptor family includes 7 members (5-HT1–5-HT7) comprising a total of 14 subtypes, making it one of the most complex families of neurotransmitter receptors [20]. Of the 14 5-HT receptor subtypes, 13, with the exception of 5-HT3, which is a ligand-gated ion channel, are classical metabotropic G-protein-coupled receptors that couple to canonical signaling pathways and elicit the expected second messenger cascades [21]. They can regulate the release of 5-HT by activating second messengers that, in turn, affect the competitive behavior of crustaceans [22]. The 5-HT is synthesized centrally in the brain, and its precursor, 5-hydroxytryptophan, is synthesized by the catalytic action of 5-hydroxytryptophan decarboxylase, which is released into the synaptic gap and regulates a wide range of behavioral actions by altering the 5-HT content in the synaptic gap and completing central nervous system (CNS: brain, subesophageal ganglion, thoracic ganglion, abdominal ganglion) signaling to metabolic G-protein-coupled and ionized ligand-gated ion channel receptors [23]. The 5-HT-related receptors are involved in the regulation of aggressive behavior. The knockout of the 5-HT1B receptor in mice can enhance aggression [24]. In addition, the expression level of 5-HT1 receptor genes affects the aggressiveness and subordinate status of fish [25,26]. Receptor sequences related to 5-HT were found in a variety of crustaceans [27,28], and full-length 5-HT1 and 5-HT2 sequences were also cloned from *Macrobrachium rosenbergii* [29].

Mud crab (*Scylla paramamosain*), belonging to the Decapoda order, is one of the most important marine crab species for the marine crab farming industry in China and Southeast Asian countries [30]. However, due to the extremely aggressive nature of mud crabs, the high stump rate and mortality due to their aggressive behavior seriously affect their yield. Therefore, it is of great research significance and in the benefit of animal welfare to reduce fighting among mud crabs by considering their BA metabolism.

In this study, we first cloned the full-length *Sp5-HTR1* gene of *S. paramamosain* and obtained polyclonal antibodies using peptide coupling. Then, we analyzed the tissue distribution of the *Sp5-HTR1* in *S. paramamosain*. Lastly, we evaluated the relationship between 5-HT and aggressive behavior. The purposes of this study were to understand the reaction mechanism of 5-HT aggressive behavior in *S. paramamosain*, provide reference data for analyzing the genetic mechanisms of aggressive behavior in *S. paramamosain*, and ultimately reduce the damage and death caused by aggressive crab behaviors during their culture and breeding.

## 2. Results

### 2.1. Cloning of the Full-Length cDNA of Sp5-HTR1 from S. paramamosain

The full-length cDNA sequence of *Sp5-HTR1* of 2318 bp was obtained by overlaying the EST sequence with the amplified fragments. The full-length cDNA contained a 205 bp 5′-untranslated region (UTR), an 891 bp 3′-UTR with a poly(A) tail, and a 1764 bp open reading frame (ORF) encoding a polypeptide of 587 amino acids. The molecular weight and theoretical isoelectric point predicted values were 63.36 kDa and 8.68, respectively, with seven transmembrane helix structures and a disulfide bond. The polypeptide sequence indicated a structurally stable hydrophobic protein (Figure 1). PSORT II analysis showed that the protein was mainly located in the endoplasmic reticulum (44.4%) and plasma membrane (33.3%). Analysis using software SignaIP 5.0 revealed no signal peptide in this sequence, and the residues constituted signature motif “7tm_1” located at amino acids 207 to 558. The secondary structure prediction results show that Sp5-HTR1 consisted of 193 α-helices, 16 β-turns, 85 extended strands, and 293 random coils, accounting for 32.88%, 14.48%, 2.73%, and 49.91% of the total structure, respectively. 

The derived *Sp5-HTR1* was similar to the *5-HTR1* of other crustaceans, with the highest similarity of 98.5% to the *5-HTR1* of *Portunus trituberculatus*, followed by 84.2% and 81.1% similarities to those of *Cancer borealis* and *Procambarus clarkii*, respectively (Figure 2A). An evolutionary developmental tree was constructed on the basis of the CRD domain of *Sp5-HTR1* and the CRD domains of *5-HTR1* in other invertebrates to reveal the relationship between *Sp5-HTR1* and the *5-HTR1* or *5-HTR1*-like peptides in other invertebrates (Figure 2B). The *5-HTR1* of *S. paramamosain* clustered first with the *5-HTR1* of *Portunus trituberculatus* and *Cancer borealis*, then with the *5-HTR1* of *Procambarus clarkii*, *Homarus americanus,* and *Panulirus interruptus*, and lastly with *Macrobrachium rosenbergii*, *Penaeus monodon,* and *Penaeus japonicus*.

### 2.2. Peptide-Coupled Antigen and Rabbit Polyclonal Antibody Analysis

We obtained 2.0 mL of a polyclonal antibody to a putative 5-HTR epitope after purification, and the antibody concentration was 1.404 mg/mL. The results for the rabbit polyclonal antibody measured using ELISA were positive and reproducible; the ELISA value was 1.416 at a dilution of 1:64,000, which was >0.3, and was thus considered valid. The primary antibody was reacted with the tissues (muscle, gill, thoracic ganglion, and hepatopancreas tissues) of the mud crab at a dilution of 1:100 and subjected to Western blotting analysis. This showed a single band on SDS-PAGE gel with a molecular weight of 63.36 kDa on the basis of the predicted molecular weight of the band between 50 and 75 kDa (Figure 3A), which was consistent with the prediction and demonstrated that the antigen was a product of *Sp5-HTR1*. The optical density values of the bands were analyzed using ImageJ software v1.8.0 to obtain the relative expression levels of Sp5-HTR1 in different tissues from the mud crab (Figure 3B). The gene was distributed in the thoracic ganglion, muscles, gills, and hepatopancreas of *S. paramamosain*, with higher expression levels observed in the thoracic ganglion and gills (*p* < 0.05), while the lowest expression was in muscle (*p* < 0.05), which is consistent with the results of other findings that 5-HT content is higher in crustacean ganglion tissue [31,32].

### 2.3. Sp5-HTR1 Response to 5-HT Injection

The mRNA and protein expression levels of *Sp5-HTR1* were altered after the 5-HT challenge. The mRNA expression level of *Sp5-HTR1* in the thoracic ganglion was significantly increased (*p* < 0.05) at 0.5 h after 5-HT injection, and was extremely significantly increased (*p* < 0.01) from 1 to 24 h after 5-HT injection (Figure 4A). When compared with the control, the mRNA expression level of *Sp5-HTR1* was significantly upregulated (*p* < 0.05) at 0.5 h in the cerebral ganglion, followed by an extremely significant upregulation (*p* < 0.01) at 2 h (Figure 4B). The mRNA expression level of *Sp5-HTR1* in muscle was highly significantly downregulated (*p* < 0.01) at 0.5 h (Figure 4C). The mRNA expression level of *Sp5-HTR1* in the gills showed highly significant downregulation (*p* < 0.01) at 4 and 12 h (Figure 4D). In addition, proteins were extracted from the thoracic and cerebral ganglion tissues of the crab at each time point after injection; then, Western blotting was performed. The Western blots show that the protein expression level of Sp5-HTR1 in the thoracic ganglion and cerebral ganglion increased gradually with time after 5-HT injection, the same as the mRNA level (Figure 4E,F).

### 2.4. Behavioral Response to Different Concentrations of 5-HT Injection in S. paramamosain

Video analysis of the mud crab behavior shows that the average moving speed and distance of the mud crabs in the 5-HT group injected with a concentration of 5 × 10^−7^ mol/L were significantly increased (*p* < 0.05, Figure 5B,C), while the average freezing time was significantly reduced (*p* < 0.05, Figure 5D). In contrast, the average moving speed, moving distance, and freezing time of the mud crabs in the 5-HT group injected with a higher concentration of 5 × 10^−3^ mol/L were not significantly different from those of the control group (*p* > 0.05, Figure 5B–D), and the average moving speed and distance were slightly lower than those of the control group. The average freezing time was slightly longer than that of the control group, possibly because the concentration of 5 × 10^−3^ mol/L 5-HT was not high enough. In addition, force transducer measurements showed that the pincer force was highly significantly reduced when crabs were injected with 5 × 10^−3^ mol/L (*p* < 0.01) and 5 × 10^−5^ mol/L (*p* < 0.05) 5-HT compared to the control group. No significant difference was found in pincer force between the mud crab group injected with 5 × 10^−7^ mol/L 5-HT and the control group (*p* > 0.05). The above experimental results show that the injection of 5-HT caused changes in the behavior and the pincer force of mud crabs, and that different concentrations of 5-HT have different effects on the behavior of mud crabs, i.e., a low concentration of 5-HT renders the mud crabs more active, while a high concentration of 5-HT inhibits their behavior.

### 2.5. Effects of Different Concentrations of 5-HT on the Aggressive Behavior of S. paramamosain

Appendix A can be seen in the experimental video of aggressive behavior of the mud crab. The exogenous 5-HT injection at different doses showed that the duration and intensity of aggression of the mud crabs gradually decreased with the increase in 5-HT concentration. The duration of aggressive behavior of mud crabs in the 5 × 10^−7^ and 5 × 10^−5^ mol/L 5-HT groups were significantly higher than those in the control group, saline group, and 5 × 10^−3^ mol/L 5-HT group (*p* < 0.05). There was no significant difference in combat duration between 5 × 10^−3^ mol/L 5-HT and 5 × 10^−5^ mol/L 5-HT groups (*p* > 0.05) (Figure 6A). Only in the 5 × 10^−7^ mol/L 5-HT group was the aggressive intensity of mud crabs concentrated in the “strong” and “moderate” states, while that of all other experimental and control groups was generally “very weak” or “weak” (Figure 6B). The above results show that the aggressive behavior of mud crabs was related to the concentration of 5-HT, and that the low concentration of 5-HT intensified the mud crabs’ aggressive behavior.

## 3. Discussion

### 3.1. Cloning and Evolutionary Analysis of Sp5-HTR1 Sequence

The 5-HT neurotransmitter is a bioactive substance that plays an important regulatory role in crustacean behavior, reproduction, molting, digestion, nutrient absorption, and metabolism [14,33]. The 5-HT1 receptors comprise five small subtypes, 5-HT1A, 5-HT1B, 5-HT1D, 5-HT1E, and 5-HT1F, which are G-coupled, inhibit adenylate cyclase, and decrease cAMP formation [34,35]. These receptors also indirectly open G-protein-gated inwardly rectifying potassium channels (GIRKs) to hyperpolarize neurons and inhibit the opening of voltage-gated calcium channels [36]. The 5-HTR cDNA sequences from several crustaceans were obtained in previous research [28,29,37]. In this study, the *5-HTR1* gene was screened according to the transcriptome of *S. paramamosain*, the full-length cDNA of *Sp5-HTR1* was cloned and sequenced, and the molecular weight of the protein was deduced (Figure 1). The peptide contained structural domain “7tm_1” without additional Class I domains, seven transmembrane helices, and one disulfide bond. *Sp5-HTR1* is a standard G-protein-coupled receptor, as it contains the core of the consensus structure without additional structural domains. *Sp5-HTR1* had high homology with the 5-HTR of several crustacean species in the GenBank database (Figure 2A), with 98.5–81.1% identity. Phylogenetic tree analysis (Figure 2B) shows the clustering of *S. paramamosain* with other crustaceans. In crustaceans, the role of 5-HT in feeding and reproduction is well-studied. However, relatively few studies have been conducted on the role of 5-HT in crustacean aggression.

### 3.2. Sp5-HTR1 Is Involved in the Regulation of Competitive Behavior of S. paramamosain

Genetic factors are one of the key influences on competitive behavior in crustaceans. The 5-HT is a highly conserved neurotransmitter in vertebrates and invertebrates, and 5-HT-related receptors are also highly conserved in crustaceans [38,39]. The activation of 5-HT receptor subtypes modulates social behaviors, especially in mammals, birds, and reptiles [40,41,42]. However, it is unclear whether the specific behavioral functions of these receptors vary across taxa or whether they correspond to changes in the circulatory system. Studies on the 5-HT receptor modulation of animal aggression focused on 5-HT1 and 5-HT2 receptors [16]. In terms of regulating the postsynaptic inhibitory effect of 5-HT on aggressive behavior, 5-HT1A and 5-HT1B receptors may have different effects in specific brain regions [43,44,45,46,47,48]. The 5-HT receptors exist in crustaceans and are involved in the regulation of their aggressive behavior [11]. Since 5-HT receptors comprise multiple types, and different receptors have different expression mechanisms, their regulatory mechanisms are still unclear [49]. The expression of the *Sp5-HTR1* gene was significantly different in different tissues (Figure 3). The concentrations of *Sp5-HTR1* in the ganglion and gills were higher than those in the muscles or hepatopancreas, which is consistent with the results of a previous study on prawns [50]. This may be because crustaceans possess an open circulatory system, and 5-HT flows through the blood circulation in gill tissues, resulting in a higher expression of the *SP5-HTR1* gene in gills [51]. After 5-HT injection, 5-HTR1 expression was significantly altered in the cerebral and thoracic ganglia, and the muscle, but not in gill tissue (Figure 4). This neurotransmitter may act through the muscle and nervous systems. Aggressive behavior is regulated by the nervous system, while muscles that are directly involved in the occurrence of aggressive behavior and gill tissue may only be associated with 5-HT transport circulation. *Sp5-HTR1* plays different roles in different tissues. In short, the functions of different tissues in agonistic behavior have not been studied in crustaceans, and this aspect needs to be further investigated.

### 3.3. Effect of Exogenous Injection of 5-HT on S. paramamosain

Crabs injected with 5-HT at a concentration of 5 × 10^−7^ mol/L showed stronger activity than that of the control group, while mud crabs in the 5 × 10^−3^ mol/L 5-HT group showed lower activity than that in the control group, although the differences were not significant. These results suggest that the regulation of 5-HT on behavior is related to its concentration. A low concentration of 5-HT promoted the aggressive behavior of the mud crabs, while a high concentration of 5-HT inhibited their aggressive behavior (Figure 5B–D). In addition, anxiety behavior, represented by movement around the container near the walls, was observed in the NS group, while more exploration of the central area of the container was observed in the mud crabs of the low-5-HT-concentration group, indicating contribution of 5-HT to overcoming anxiety (Figure 5A). The struggle among male crabs of the low concentration 5-HT group was more intense and longer than that of the control and high-5-HT-concentration groups (Figure 6). Other studies found that the content of 5-HT in crustaceans increased after fighting [49,52], and that the exogenous injection of 5-HT led to an increase in aggressive behavior [53,54,55], which is consistent with the results of this experiment. Thus, the level of 5-HT in crustaceans is closely related to their competitive behavior, and an increase in 5-HT level reduces crustacean aggression. Some results in crayfish suggest that 5-HT has the opposite effect on aggression against smaller or larger competitors, as 5-HT-injected crayfish showed increased levels of aggression against larger competitors, and decreased levels of aggression against smaller competitors. This suggests that 5-HT acts on the integration of the brain centers of risk assessment and decision making rather than on the aggressive response [56]. There are different receptor subtypes in invertebrates, and their pharmacology, affinity for 5-HT, target location, and role in the nervous system may influence the role that 5-HT plays in invertebrate behavior. The effect of 5-HT injection on the behavior of *S. paramamosain* thus needs further study.

Therefore, from the results of this experiment, 5-HT, as a neurotransmitter, participates in the aggressive behavior of the mud crabs, promotes aggressive behavior at low concentrations, and inhibits aggressive behavior at high concentrations. If the content of 5-HT could be increased in mud crabs as part of their commercial production, the occurrence of fighting behavior could be reduced, which would improve production. Additives of L-tryptophan, the precursor of 5-HT, to farmed crustaceans increased the content of 5-HT in their hemolymph, inhibited the occurrence of fighting behavior, and improved the survival rate [57,58,59].

## 4. Materials and Methods

### 4.1. Animal Collection and Maintenance

Mud crabs (*S. paramamosain*; 20 ± 3 g) were collected from a farm in Sanmen, Zhejiang province, China, and were maintained in separate tanks of dimensions 20 cm × 15 cm × 10 cm (length × width × height) for at least seven days under single rearing conditions before the experiments. The tanks were filled with thoroughly aerated seawater to a depth of 8 cm. Intact crabs were reared individually to avoid social contact, as isolation can also increase the animal’s aggressiveness [57]. The crabs were cultured at 19 and 21 °C, and maintained in seawater (salinity 25 ± 1), one-third of which was changed daily. The crabs were fed clams once daily from 18:00 to 21:00, and feeding was stopped a night before the experiment. Injured crabs were not used for this study.

### 4.2. Rapid Amplification of cDNA Ends

Total RNA was extracted using a TRIzol^®^ Plus RNA Purification Kit (Invitrogen, Carlsbad, CA, USA) followed by the reverse transcription of 2 μg of total RNA using SuperScript™ III First-Strand Synthesis SuperMix (Invitrogen, Carlsbad, CA, USA) for qRT-PCR to synthesize the cDNA.

A partial sequence of *Sp5-HTR1* was obtained from the transcriptome sequencing data (GenBank accession no: OP985334) of *S. paramamosain*. Full-length cDNA was obtained using expressed sequence tag (EST) analysis and rapid amplification of cDNA ends (RACE) techniques. The 5′- and 3′- end sequence amplification gene-specific primers are listed in Table 1. PCR (Platinum^®^ PCR SuperMix, High Fidelity, Invitrogen, Waltham, MA, USA) was performed using a 50 μL reaction volume according to the manufacturer’s instructions. The PCR product was isolated on 1.5% agarose gels and purified with a PCR purification kit (BBI Company, Halifax, NS, Canada). The PCR product was cloned into a pGM-T simple vector (Tiangen Biotech, Beijing, China) and transformed into highly efficient DH5α chemoreceptor cells. The positive recombinants were selected using antikanamycin selection before PCR screening with the primers m5-HT-1-F and m5-HT-1-R. Three positive clones were confirmed for sequencing (BGI Company, Shenzhen, China). The full sequence of *Sp5-HTR1* cDNA was assembled using the Vector NTI software 11.5.3 after removing the vector sequences.

### 4.3. Sequence Analysis

The cloned sequence and deduced amino acid sequence of *Sp5-HTR1* were analyzed with other 5-HT receptor sequences for identity and similarity by using BLAST from the National Center for Biotechnology Information (www.ncbi.nlm.nih.gov/BLAST/, accessed on 7 December 2022). The protein sequence of 5-HTR1 was predicted by Expasy (http://web.expasy.org/protparam/, accessed on 7 December 2022), and its physicochemical properties were analyzed. PSORT II prediction (http://psort.hgc.jp/form2.html/, accessed on 7 December 2022) was used to predict subcellular localization. Analysis of the transmembrane structure of 5-HTR1 protein was performed using TMHMM 2.0 (http://www.cbs.dtu.dk/services/TMHMM/, accessed on 7 December 2022). The simple modular architecture was identified using SMART 4.0 to predict the presence and location of the protein domain (http://smart.embl-heidelberg.de/, accessed on 7 December 2022). Prediction of the presence of signal peptides was performed by SignalP (http://www.cbs.dtu.dk/services/SignalP/, accessed on 7 December 2022). The secondary structure analysis was performed using sopma (https://npsa-prabi.ibcp.fr/cgi-bin/npsa_automat.pl?page=/NPSA/npsa_sopma.html/, accessed on 7 December 2022). Sequence alignment of *Sp5-HTR1* with those of other animals was performed with ClustalX software 1.83 and the Multiple Alignment program (http://www.biosoft.net/sms/, accessed on 7 December 2022). A phylogenetic tree was constructed using the neighbor-joining (NJ) algorithm with MEGA software 11.0.11.

### 4.4. Peptide-Coupled Antigen and Preparation of the Antibody

Synthetic polypeptide sequences (GRENTTSDDWNYTLC) were coupled to carrier proteins keyhole limpet hemocyanin (KLH) and bovine serum albumin (BSA). The polypeptide coupled to the KLH protein was used to immunize rabbits, and the BSA-conjugated complex was used for screening to exclude antibody reflections against KLH. The Sp5-HTR1 protein was purified and used as an antigen to immunize rabbits for polyclonal antibodies prepared by Hangzhou HuaAn Biotechnology. The antibody titers were determined using the enzyme-linked immunosorbent assay (ELISA) method.

### 4.5. Sp5-HTR1 Expression in the Tissues of S. paramamosain

In this experiment, crabs were selected at random, and divided into a control group (CK), a normal saline group (NS), and an experimental group (5-HT). The experimental group was injected with 5 mg/mL of 5-HT, 20 µL per crab; the normal saline group was injected with the same volume of normal saline; and the control group was not injected. The mRNA expression levels of the *Sp5-HTR1* gene were separately analyzed at seven time points (0–24 h) after injection of 5-HT into the tissues. Muscle, gill, thoracic ganglia, and hepatopancreas tissues were taken for RNA extraction, with three replicates each. All tissue samples were stored at −80 °C until use.

Quantitative RT-PCR assay with Power SYBR^®^ Green PCR Master Mix was used to determine the distribution of *Sp5-HTR1* in the tissues of *S. paramamosain*. A 141 bp fragment of *Sp5-HTR1* was amplified using the primers 5-HT1-F and 5-HT1-R (Table 1). Primers for GAPDH (GAPDH-F and GAPDH-R) were used as the internal controls. RT-PCR was performed using a final volume of 20 μL containing 3 μL of PCR-grade water, 10 μL of 2× master mix, 1 μL of each primer (10 mmol L^−1^) for GAPDH or *Sp5-HTR1*, and 5 μL of the cDNA mix. The relative expression levels of *Sp5-HTR1* were calculated using the 2^−ΔΔCt^ method [60]. 

### 4.6. Western Blotting

Tissue sampling for protein extraction was the same for the gene expression analysis. Total protein extracts were used to detect Sp5-HTR1 using the Western blot technique. Tissues were independently homogenized on ice in T-PER Tissue Protein Extraction Reagent (Thermo Scientific, Waltham, MA, USA) and then centrifuged at 10,000× *g* for 5 min. The supernatant with an equal volume of added loading buffer was stored at −20 °C until use. Western blotting was performed with a 15% separating gel and a 5% stacking gel. The proteins were transferred from the gel to PVDF membranes. The membranes were washed three times in PBST and blocked with blocking buffer (1% casein) for 2 h. Then, the PVDF membranes were incubated with primary and secondary antibodies at 37 °C for 1 h. Using SuperSignal^®^ West Dura Extended Duration Substrate, the laboratory-grade X-ray films were exposed [61]. GAPDH was selected as the control, and the expression level of Sp5-HTR1 was analyzed using Image J software.

### 4.7. Effect of 5-HT on the Behavior of S. paramamosain after Injection

The concentrations for the experimental group were 5 × 10^−3^, 5 × 10^−5^, and 5 × 10^−7^ mol/L 5-HT. The solution (20 μL for each crab) was injected into the joint membrane of the crab foot, with nine replicates each. The injection dose and method were determined according to a previous study [11]. The normal saline group (NS) was injected with the same amount of normal saline, and the control group (CK) was not injected. Exogenous 5-HT can rapidly be distributed throughout the hemolymph after injection, and most of it can be absorbed by tissues within five minutes [62]. The injected crab was placed in the center of an open experimental water tank of dimensions 30 cm × 30 cm × 20 cm (length × width × height). Each tank had 5 cm of seawater and an opaque water pipe. After 10 min of adaptation, the behavioral trajectory was observed and recorded with EthoVision XT 12.0 (Noldus, Wageningen, The Netherlands) for 30 min. At the end of each experiment, seawater was removed, and the tank was refilled with new seawater.

The changes in the pincer force of crabs treated in the same groups were measured [63]. Healthy male crabs with basically the same specifications and complete limbs were selected to measure the pincer force without injection, after the injection of normal saline, and after the injection of 5-HT to analyze the effect of 5-HT on the change in pincer force.

### 4.8. Observation of Aggressive Behaviors

The struggle among crabs of the same sex was more intense than that among crabs of the opposite sex, and the struggle among male crabs was more intense than that among female crabs [11,64]. Therefore, we used male × male paired fights. Three experimental groups with treatments of 5 × 10^−3^, 5 × 10^−5^, and 5 × 10^−7^ mol/L 5-HT crabs were set up, with three crab pairs in each group, and the two crabs in each pair were the same size. For the control, the NS group was injected with normal saline, and the CK group was not injected. Each pair was maintained in a new tank of dimensions 25 cm × 25 cm × 20 cm (length × width × height). Each tank had 5 cm of seawater and was divided into equal halves with an opaque partition. Two crabs were placed on either side of the partition. After 30 min, the partition board was removed for recording the agonistic behaviors of the crabs with EthoVision XT for 30 min. This length of observation exceeded the actual duration of aggressive behavior, ensuring that the complete process of aggressive behavior could be recorded. Referring to previous research [52], the aggressive behaviors were divided into two categories: non-contact behaviors (move to, move away, and cheliped display) and contact behaviors (push, strike, and climb on) (Table 2). The duration of aggression was the time between the emergence of aggressive behavior by an individual and the emergence of successive withdrawal behavior by one party. According to previous studies [52,65], we defined the intensity of aggressive behavior as follows: (1) very weak—one individual approaches another and shows aggressive behavior, while the other individual shows withdrawal or obedience without physical contact; (2) weak—physical contact occurs, including pushing and shoving, but no attempt is made to hold or fight; (3) moderate—the fight between two individuals escalates, and there are striking and grasping behaviors; and (4) strong—individuals use their chelipeds to tear and grasp various parts of each other’s body and attempt to damage or remove each other’s appendages, with the loser showing retreating behavior but the loser unfolding their chelipeds to constantly demonstrate and restart the fight.

### 4.9. Statistical Analysis

The data are expressed as the mean ± standard deviation, and one-way analysis of variance was used for comparisons among the groups, which was conducted with SPSS 26.0 software (SPSS Inc., Chicago, IL, USA). GraphPad Prism 8.0 (GraphPad Software Inc., La Jolla, CA, USA) was used for mapping. A *p* value < 0.05 indicates a statistically significant difference, and *p* < 0.01 indicates that the difference was extremely significant.

## 5. Conclusions

In this study, we identified a previously undescribed *5-HTR* and described the impact of *Sp5-HTR1* on aggressive behavior through 5-HT metabolism in *S. paramamosain*. In addition, we clarified the metabolic relationship between 5-HT and *Sp5-HTR1* to determine whether *Sp5-HTR1* could regulate synaptic 5-HT metabolism and act on the aggressive behavior of *S. paramamosain*, and showed that the aggressive behavior of *S. paramamosain* was regulated by 5-HT content. We speculate that *Sp5-HTR1* acts on the nervous system to produce important changes in the behavior of *S. paramamosain,* and that these changes are closely related to aggressive behavior.

## Figures and Tables

**Figure 1 ijms-24-04211-f001:**
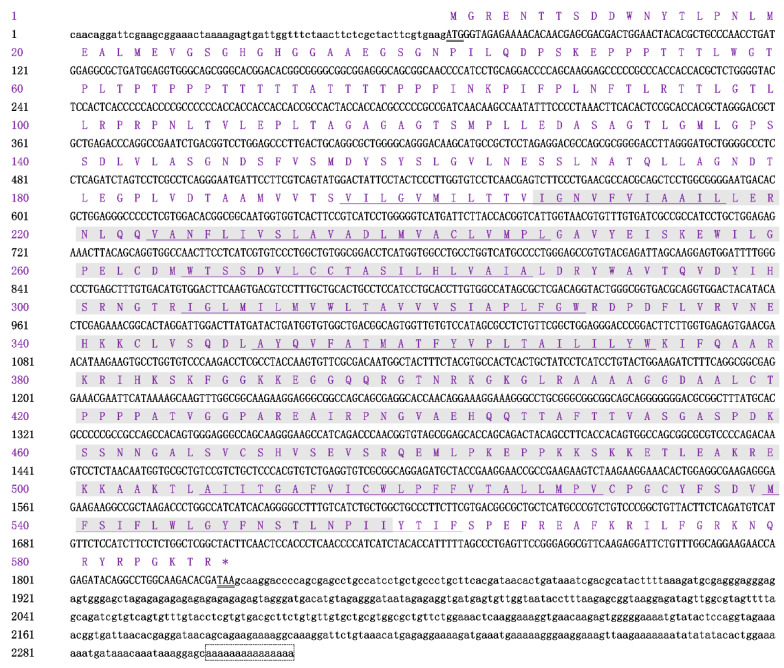
Nucleotide and deduced amino acid sequences of the *Sp5-HTR1* gene. The amino acid sequence is shown as single letters underneath the encoding nucleotide sequence. The numbers on the right indicate nucleotide and amino acid numbers. The “7tm_1” domain is shaded in gray. The initiator codon and termination codon are double-underlined. The “*” is represented as a termination codon. The seven transmembrane regions are underlined. Dashed boxes indicate the poly (A) tail.

**Figure 2 ijms-24-04211-f002:**
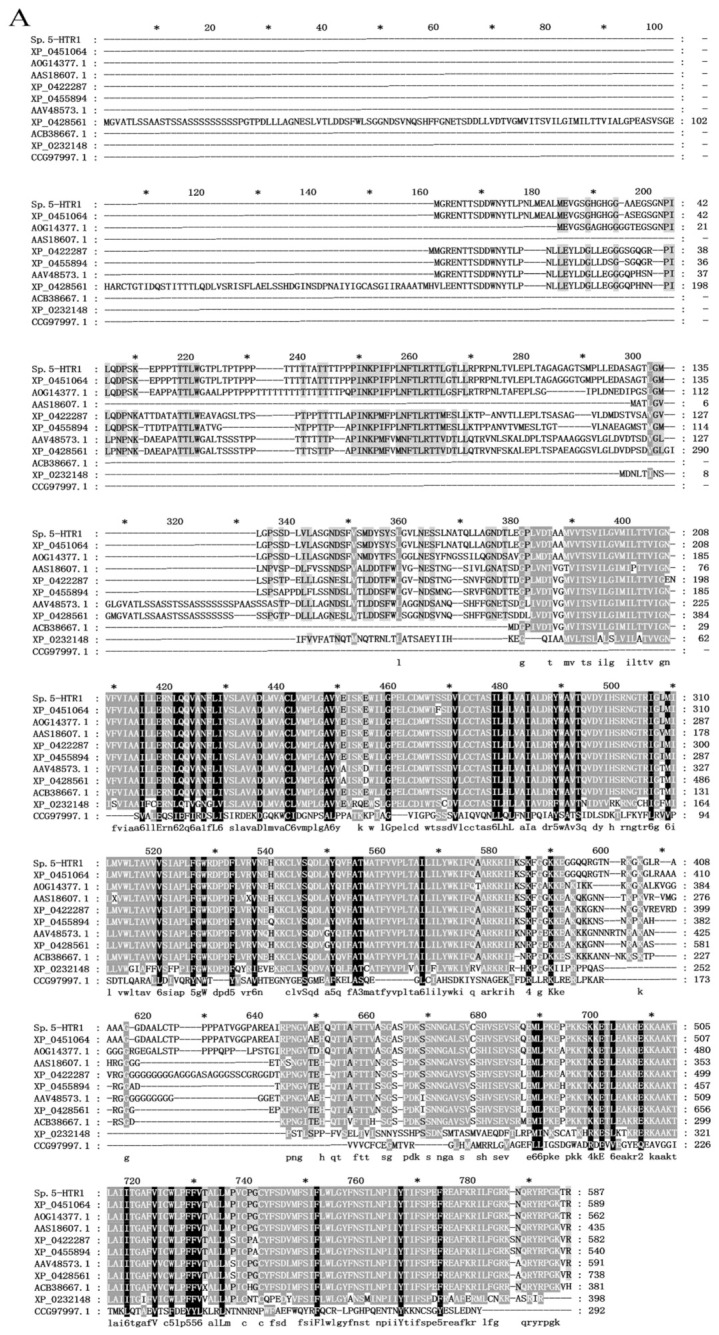
(**A**) Multiple sequence alignment and (**B**) phylogenetic tree of CRD in *Sp5-HTR1* and other *5-HTRs*. The conserved amino acid residues are shaded in black. Sequences: 5-hydroxytryptamine receptor-like (*Portunus trituberculatus*; GenBank accession no: XP_045106432.1); serotonin receptor type 1A (*Cancer borealis*; GenBank accession no: AOG14377.1); 5-hydroxytryptamine receptor-like (*Procambarus clarkii*; GenBank accession no: XP_045589401.1); 5-hydroxytryptamine receptor-like (*Homarus americanus*; GenBank accession no: XP_042228751.1); 5-HT1 receptor (*Penaeus monodon*; GenBank accession no: AAV48573.1); type 1 serotonin receptor (*Panulirus interruptus*; GenBank accession no: AAS18607.1); 5-hydroxytryptamine receptor-like (*Penaeus japonicus*; GenBank accession no: XP_042856161.1); type 1 serotonin receptor (*Macrobrachium rosenbergii*; GenBank accession no: ACB38667.1); 5-hydroxytryptamine receptor 2A-like isoform X2 (*Centruroides sculpturatus*; GenBank accession no: XP_023214834.1); glutamate receptor, metabotropic 1, partial (*Carassius auratus*; GenBank accession no: CCG97997.1); 5-hydroxytryptamine receptor 1F (*Homo sapiens*; GenBank accession no: NP_000857.1); 5-hydroxytryptamine receptor 1A (*Rattus norvegicus*; GenBank accession no: NP_036717.1); serotonergic G-protein-coupled receptor (*Echinococcus canadensis*; GenBank accession no: UDM84210.1); and 5-HT1 receptor (*Pinctada fucata*; GenBank accession no: AIW04132.1).

**Figure 3 ijms-24-04211-f003:**
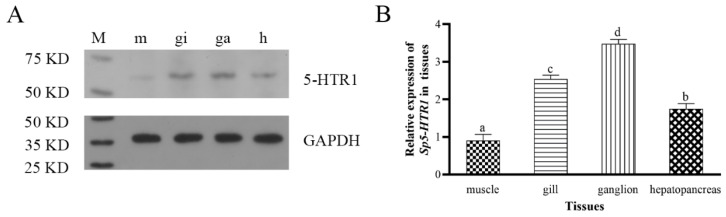
(**A**) Relative expression levels of Sp5-HTR1 in the tissues of *S. paramamosain*. GAPDH serves as an internal standard. (**B**) Bars correspond to the standard deviation (SD) of the mean values (*n* = 3). Significant differences (*p* < 0.05) between different tissues are indicated with different letters. Abbreviations: M, marker; m, muscle; gi, gill; ga, thoracic ganglion; h, hepatopancreas.

**Figure 4 ijms-24-04211-f004:**
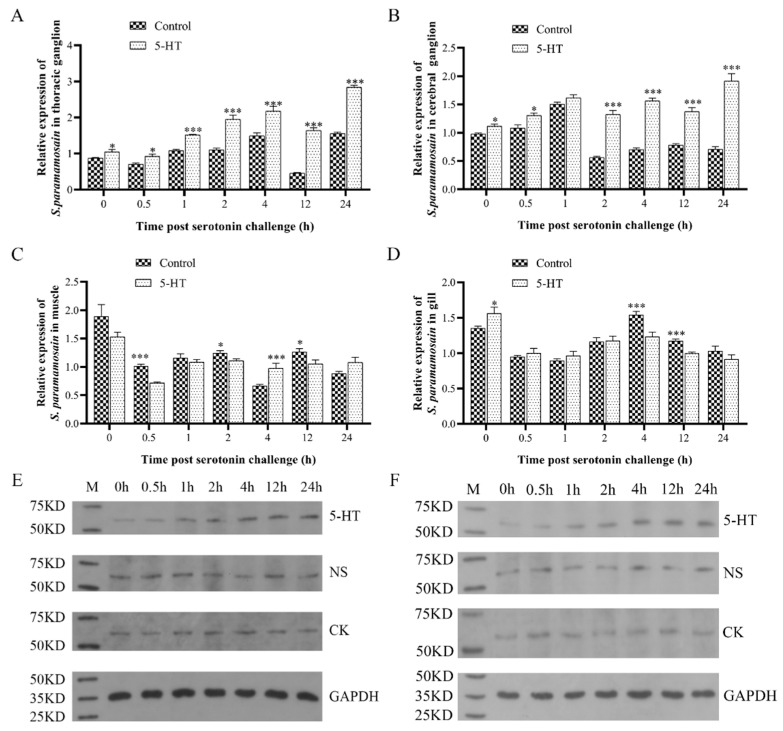
*Sp5-HTR1* mRNA expression levels in the (**A**) thoracic ganglion, (**B**) cerebral ganglion, (**C**) muscle, and (**D**) gill of *S. paramamosain* injected with 5-HT. *Sp5-HTR1* protein expression levels in the (**E**) thoracic ganglion and (**F**) cerebral ganglion of *S. paramamosain* injected with 5-HT. (**A**–**D**) Bars correspond to the standard deviation (SD) of the mean values (*n* = 3). Asterisks indicate significant differences. * *p* < 0.05; *** *p* < 0.01. Abbreviations: CK, control check group; NS, normal saline group; GAPDH, internal standard.

**Figure 5 ijms-24-04211-f005:**
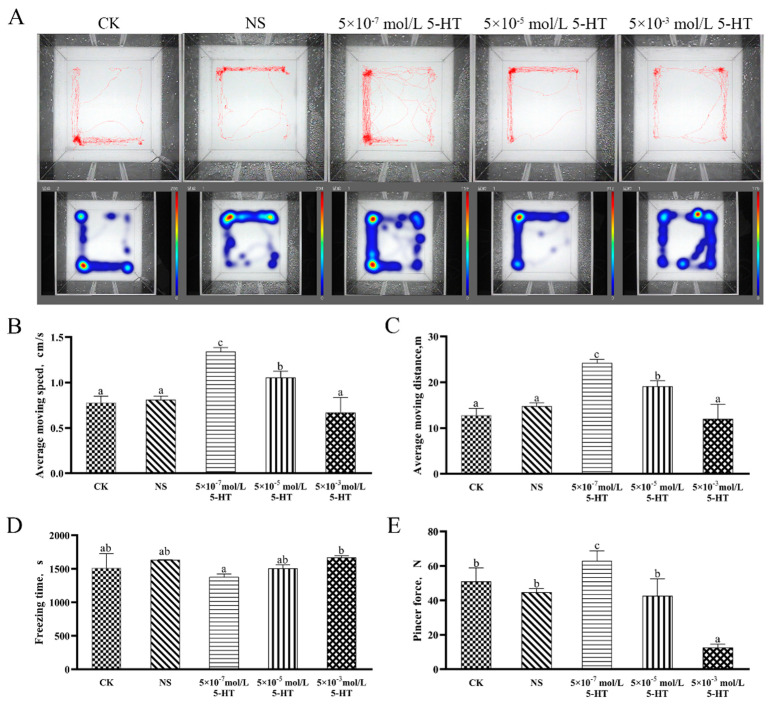
(**A**) Trajectory of mud crabs injected with normal saline and different concentrations of 5-HT normal saline. *S. paramamosain* measures of (**B**) average moving distance, (**C**) average moving speed, (**D**) freezing time, and (**E**) pincer force. (**B**–**E**) Bars correspond to the standard deviation (SD) of the mean values (*n* = 9). Significant differences (*p* < 0.05) between the challenged and control groups are indicated with different letters. Abbreviations: CK, control check group; NS, normal saline group.

**Figure 6 ijms-24-04211-f006:**
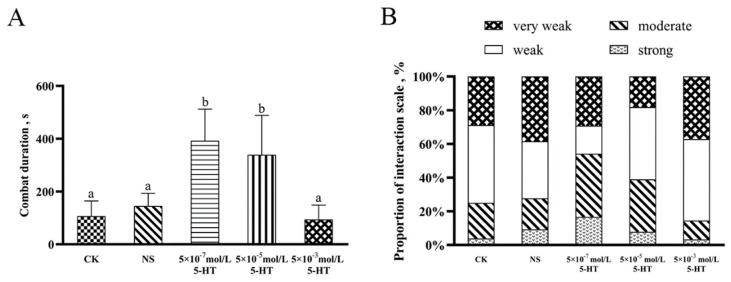
(**A**) Combat duration and (**B**) proportion of interaction scale between different concentrations of 5-HT. (**A**) Bars correspond to the standard deviation (SD) of the mean values (*n* = 3). Significant differences (*p* < 0.05) between the challenged and control groups are indicated with different letters. Abbreviations: CK, control check group; NS, normal saline group.

**Table 1 ijms-24-04211-t001:** Oligonucleotide primers used in the study.

Primer	Sequence (5′–3′)	Sequence Information
m5-HT-1-F	CTACACGCTGCCCAACCTG	RT-PCR Primer
m5-HT-1-R	GATGGAGAACATGACATCTGAGAAG	RT-PCR Prime
r5-HT-1-F1	CAAGGGAAGCCATCAGACCCAACGGTGTA	5-RACE Primer
r5-HT-1-F2	CGTCCCCAGACAAGTCCTCTAACAATGG	5-RACE Primer
r5-HT-1-R1	GGGGTGGGGGTGAGTGGAGTACC	3-RACE Primer
r5-HT-1-R2	GCTGGGGTCCTGCAGGATGGGGTTG	3-RACE Primer
5HT1-F	GAGGTGATGAGTGTTGGTAATACC	Specific primer (for real-time PCR)
5HT1-R	CCTTTCCTTGAGTTTCCAGAACAG	Specific primer (for real-time PCR)
GAPDH-F	CTAAGGCTGTAGGCAAGGTCATT	Primers for GAPDH
GAPDH-R	CCAGAATGCAAGTCAGGTCAAC	Primers for GAPDH

**Table 2 ijms-24-04211-t002:** Description of aggressive behaviors.

Behavior	Description
Move to	One crab approaches the other crab.
Move away	One crab moves away from the other crab after aggressive behaviors.
Chelipeds display	Chelipeds are held out in front.
Push	One crab uses its chelipeds to push the opponent away.
Strike	One crab strikes the opponent with its chelipeds.
Grasp	One crab uses its chelipeds to pinch the carapace, chelipeds, or legs of the opponent.
Climb on	One crab climbs on top of the other crab.

## Data Availability

The datasets generated during the current study were deposited in the National Center for Biotechnology Information (NCBI) GenBank database (https://www.ncbi.nlm.nih.gov/, accessed on 7 December 2022) under the accession number OP985334.

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
