# Peer review of "Identification and Characterization of 5-HT Receptor 1 from Scylla paramamosain: The Essential Roles of 5-HT and Its Receptor Gene during Aggressive Behavior in Crab Species"

_ijms, 2023, doi:10.3390/ijms24044211_

Round 1

Reviewer 1 Report

This is an interesting  contribution to understand what controls aggressive behavior in a commercial species of crab. Just made a few suggestions in the attached file.

Reviewer 2 Report

Letter to Authors
ijms-2118475-v1
Identification and characterization of 5-HT receptor 1 from Scylla paramamosain: The essential roles of 5-HT and its receptor gene during aggressive behavior in crab species
Xinlian Huang, Yuanyuan Fu, Wei Zhai, Xiaopeng Wang, Yueyue Zhou, Lei Liu, Chunlin Wang

220110

Dear Authors,
I am sorry to say your MS has importance only on the material species. Such localized study is good for publishing in a local journal or more specified media dealing with crustacean biology or aquaculture. I suggested the editor to return your MS this time. As further hints when submitting a suitable journal, I made several point-to-point comments below for improvement. Words in braces indicate options, and bracketed words can be omitted. It is a pleasure of me if below are helpful for submitting to another suitable journal.

L11
the aggressive behavior of crustaceans -> aggressive behavior of the crustaceans

L13,23,28
behavior -> behaviors

L15
Reverse Transcription-Polymerase Chain Reaction (not a proper noun) -> in lower case

L19
the ganglion
What ganglion?

L22
And the results indicated that after (wordy with a repeated word) -> delete

L24,25
injection -> injected

L28
mud -> delete
See L4,14.

L29 keywords
Scylla paramamosain; 5-HT; aggressive behavior -> replace
Avoid listing words which appear also in the title. Duplicate hits upon computer search do not make sense. Give words that do not appear in the title to draw attention from wider readership. Posting words that neither appear in the abstract is better, because even in full-text search/indexing robots may not weigh much on words deeper (posterior) in the text. Hint: biogenic amines, serotonin, 5-hydroxytryptamine, neurotransmitter, Portunidae, Decapoda, etc.

L32
behaviors (grammar) -> behavior

L35
rights to control key (does not make sense) -> access to

L37
powerful (does not make sense) -> armored
tentacles, and the occurrence -> tentacles. The occurrence (break sentence here)

L41
light[4-6] -> light [4-6] (insert a white space)

L50-95
This story flow is not good. A paragraph of L50-64 talks about 5-HT and that of L65-95 about its receptor. Each includes general, crustacean, and S. paramamosain specific matters, and hence the story goes back-and-forth. I suggest you to make three paragraphs, telling (1) general matters of 5-HT and (2) its receptors (responsive cascades), and (3) crustacean and S. paramamosain specific matters.

L53
aggressive -> {aggressiveness, aggressive behavior}

L55-60
Some researchers have used .. combat durability [18]. -> revise
Do not make a simple reference list (A stated this, B analyzed that, C argued it, or alike). Simple reference lists bloat your MS, dilute your originality, and undersell your own research. Use noun phrases to make abstract contents of those references. Hint: "induction of {agonistic, aggressive} behaviors} by external 5-HT intake", "in {crustaceans, lobster and crayfish}", etc. Particular decapod species names are not essential.

L62,95
S. paramamosain ?
If your study is important only for this species, this MS is not suitable to a highly impacted international journal. See L28,36,etc.

L79-86
Previous studies have shown .. in highly aggressive individuals [26].  -> revise
Do not make a simple reference list. Fish species names are not essential.

L86-92
In crustaceans, .. Macrobrachium rosenbergii [29]. -> revise
Do not make a simple reference list.

L105
Sp5-HTR1 -> in Roman
S. paramamosain -> in Roman
Romanized (upright) words in a fully Italicized line are equivalent to Italics in a Roman line. This is the formal scripting.

L106
Sp5-HTR1 -> in Italics

L108
3'-untranslated region (UTR) -> 3'-UTR

L109
acids with .. a disulfide bond. (long sentence) -> acids. Its predicted values of molecular weight and theoretical isoelectric point (pI) were respectively 63.36 KD and 8.68 with seven transmembrane helix structures and a disulfide bond. (break sentence)

L120
Portunus trituberculatus -> in Italics

L121
Cancer borealis -> in Italics
Procambarus clarkii -> in Italics

L125,131,185 figure pictures
I ask you to improve your figure resolution quality. The current delivery is hardly suitable for printing. Readers will feel difficulty to see the figures for evaluation.
To improve the figure picture resolution, choose either of the following ways.

The best way:
Original picture drawn by X application
 -> export to a pdf file, or copy to clipboard
 -> open it with Illustrator, or paste onto an Illustrator picture
 -> omit unwanted picture frame or canvas hemming when necessary
 -> select all
 -> slowly drag-and-drop onto a word document
This makes the picture with vector data and searchable text.

2nd best:
Original picture drawn by X application
 -> export to a pdf file
 -> open it with Photoshop or GIMP upon an appropriate resolution to make it full-HD or larger size
 -> trim margins when necessary
 -> export to a png file
 -> paste it onto a word document

3rd best:
Original picture drawn by X application
 -> enlarge to full-HD or larger size on screen
 -> print-screen and paste it onto a Photoshop or GIMP picture canvas
 -> combine pictures
 -> export to a png file
 -> paste it onto a word document

IMPORTANT NOTES
You might be surprised, but PPT/XLS is hardly compatible with Word! Exact reproduction of PPT/XLS pictures in fine resolution is impossible unless mediated through pdf.
Do not use jpeg format that deceives human eyes with unwanted blobs on white background just like the current delivery (enlarge to, say, 400% to see it). Likewise, do not use a jpeg compression option when printing to a pdf file. Use a zip compression option instead, if available.
Use print quality option when exporting a word document to a pdf file.

L125 figure 1 picture
In order to save spaces, Line width of the sequence should be doubled or more (=> 120 bp).

L131 figure 2 picture
To be reader-friendly, enlarge the picture. Especially for the sub-figure B, font size is too small to see. It should be at least in the same size as those in the main text.

L149
polyclonal antibody -> polyclonal antibody to a putative 5-HTR epitope (reader-friendly)

L153
What is the cave green grab?

L159
The results showed that (verbose) -> delete

L160,166,187,215,
S. paramamosain -> in Italics

L160,161,169
Which ganglion? Does the word "ganglia" (L160,161) mean two ganglia from both brain and thorax? That in L164 is OK because of citing matters.

L164
tissues[30,31] -> tissues [30,31] (insert a white space)

L166
Insert "GAPDH serves as an internal standard."
You have already mentioned this in the M & M section, but because of the order of appearance, you would better be reader-friendly here.
bars -> bars (B)

L180
brain ganglion -> thoracic and cereberal ganglion ?
See L188.

L183
cerebral -> cereberal

L185 E & F
5-HT -> 5-HTR1 ?

L188
Insert "NS, injection of saline; CK, no injection; GAPDH, internal standard."
Or, see L218.

L189
bars -> bars (A-D)

L191,219,271
S. paramamosain -> in Roman

L192-193
EthoVision XT .. The results (redundant) -> Video analysis of [the mud crab] behavior

L196,197,201
freezing -> pausing ?

L198
a concentration of -> a higher concentration at (reader-friendly)

L202
the concentration
Which concentration?

L205
in the mud crab group (wordy) -> when

L206
with (repeated word) -> to

L207
found between the pincer force of (does not make sense) -> found in the pincer force between

L213 figure 5 picture
NS and 5x10^-3 groups appear similar in freezing (pausing) times (D).
Ck and 5x10^-7 groups seems showing similar pincer force (E).
NS and 5x10^-5 groups appear similar in pincer force (E).
Is it OK?

L220-221
The effects of exogenous .. in the figure below illustrating (wordy) -> Exogenous 5-HT injection at different doses showed

L223
The results showed that (verbose) -> delete

L224
10-7 -> 10^-7
10-5 -> 10^-5

L225
10-3 -> 10^-3

L227
10^-3 mol/l 5-HT groups -> 10^-3 mol/l 5-HT and control groups

L234 figure picture
Patterns on the bars in B stand for "very weak" and "strong" are difficult to distinguish. Use different patterns or colors which can clearly be seen.

L236
bars -> bars (A)

L241
SP5-HTR1 -> Sp5-HTR1 (in Roman)

L248
channels[35] -> channels [35]

L248-250
In crustaceans, .. Eriocheir sinensis [36]. (simple reference list) -> cDNA sequences of 5-HTR have been obtained from several crustaceans [28,29,36].

L252
to obtain its nucleotide sequence (verbose) -> delete

L253
to be 63.36 KDa and encoded 587 amino acids -> delete
Do not make simple repetition of your results in the discussion section. Put then-what matters forward. Use noun phrases to make abstract your results.

L257
The results of (verbose) -> delete

L259-261
several species .. respectively. (simple reference list) -> several crustacean species in the GenBank database (Fig. 2A) with 98.5-81.1% {identity, match}.

L261
A comparative analysis .. conserved in crustaceans. (redundant) -> delete

L262
Based on the results of the 5-HTR1 amino acid sequence alignment, (verbose) -> delete

L263
using MEGA X software (}redundant) -> delete

L264-267
that the 5-HTR1 of the S. paramamosain clustered .. and Penaeus japonicus. -> move to the result section (around L124).
Do not make simple repetition of your results.

L267-268
These species are .. thus the results were (verbose) -> clustering of S. paramamosain with other crustaceans
Use noun phrases to make abstract your results.

L278-296
Studies on 5-HT receptor .. in specific brain regions. (simple reference list) -> delete

L302
Sun's research results -> results in a prawn
What is more informative than who.

L305
cranial ganglia, thoracic ganglia -> cranial and thoracic ganglia

L306
(Fig. 4), suggesting that it -> (Fig. 4). It (break sentence here)

L307
nervous system and that aggressive behavior is regulated -> nervous system. Aggressive behavior is regulated (break sentence here)
Connection of behavior to nervous system is so clear that neither "may be" nor "suggested" is necessary.

L308
gill tissue is -> gill tissue may be

L313-337
As a neuromodulator, 5-HT .. CHH release. (simple reference list) -> delete
Emphasizing cAMP-PKA signaling pathway, CHH, etc undersell your research without experimental results on these matters.

L337
In the present study, 5-HT was injected into mud crabs, and -> delete

L338-339
significantly higher .. reduced freezing time, and (repetition of results) -> delete
Use noun phrases to make abstract your results.

L340,341
group -> groups
Both NS and CK worked as negative controls.

L341
movement speed and distance -> activity
Make abstract your results.
those in -> delete

L344,346
of mud crabs (redundant) -> delete

L346-349
we found that .. This suggested (repetition of results) -> Anxiety behavior represented by wall attachment movement around the container in NS group and more exploration in the central area of the container observed in the low concentration 5-HT group of the mud crab
Use noun phrases to make abstract your results.

L350
on mud crabs (redundant) -> delete

L350-352
In another study, .. aggression. (redundant) -> delete

L352
between crabs -> between male crabs

L353
control group -> control groups

L355-363
In other studies, .. in the circulatory system [68]. (simple reference list) -> delete
You may cite references at the end of next sentence.

L366
on crayfish aggression (repeated) -> on aggression

L382 M & M
Description of sampling scheme is duplicated. Disposition of duplicate items is needed.

L393
Crabs -> To examine Sp5-HTR1 expression responding to the ligant, crabs

L394
5 μg/g of 5-HT ?
Volume (conc.)?

L395,448
amount -> volume ?

L408
PCR was performed using a 50 μL reaction volume according to the manufacturer's instructions. ?
Enzyme? Kit? PCR profile?

L418 sequence analysis
References of softwares are necessary except for very widely used BLAST.

L445
5-HT solution ?
Volume?

L447
previous studies [11] -> a previous study [11]
You are citing one paper.

L449
Studies have shown that -> delete

L462
According to the research results of Yang and Wu [11,73], -> delete
References may be cited at the end.

L467-468
Referring to .. different concentrations; the (redundant) -> delete

L469
NS group -> For control, NS group

L474
30 minutes, a period that exceeded -> 30 minutes. This length of observation exceeded (break sentence)

L476
Sneddon's -> a previous
definition of the aggressive behaviors of mud crabs (Table 2) (redundant) -> delete
Table 2 may be cited at the end.

L481
the research results of Muramatsu and Sneddon -> previous {studies, reports}

L492-494
Four crab samples .. into the tissues.
Does "Four" indicate iteration?
yes ->
Four individual crabs were randomly selected from the same pond and were used separately for the mRNA expression level of the Sp5-HTR1 gene at seven time points (0-24 h) after injection of 5-HT into the tissues. Muscle, gill, cereberal and thoracic ganglia, and hepatopancreas tissues were taken for RNA extraction.

L497
Primers for GAPDH (internal standard) and GAPDH-F and GAPDH-R (does not make sense) -> Primers for GAPDH (GAPDH-F and GAPDH-R)

L500
β-actin ? -> GAPDH ?

L505-506
Total proteins .. were used to (duplicate description) -> Tissue sampling for protein extraction was the same for the gene expression analysis. Total proteins extracted were used to

L507,516
Sp5-HTR1 (protein) -> in Roman

L507
The muscle, .. were homogenized -> Tissues were [independently] homogenized

L510
loading buffer ?
Does it have a common name?

L511
transferred
Common name of the transferred buffer?

L520
Supplier information of GraphPad is necessary.

L523 conclusions
Statements are almost totally without foundation. You did not examined 5-HT metabolism.

L548-550
This research was funded .. Ningbo University. (redundant) -> delete
You may mention about technicians, supervisors, coordinators, officers, etc who contributed to your research activities.

L552 references
Check the reference list carefully again from the beginning. Reference lists are frequently hotbeds of errors. You might add, omit or swap citation in the main text on the way internal revision. Numbering of the references might then shift. If so, readers think you are making irrelevant citation. It is the authors' responsibility that all references are properly cited.

The following items may be helpful for further discussion.

Duvaud S, Gabella C, Lisacek F, Stockinger H, Ioannidis V, Durinx C. 2021. Expasy, the Swiss Bioinformatics Resource Portal, as designed by its users. Nucleic Acids Res 49:W216-W227.

Kumar S, Stecher G, Li M, Knyaz C, Tamura K. 2018. MEGA X: Molecular evolutionary genetics analysis across computing platforms. Mol Biol Evol 35:1547-1549.

Larkin MA, Blackshields G, Brown NP, Chenna R, McGettigan PA, McWilliam H, Valentin F, Wallace IM, Wilm A, Lopez R, Thompson JD, Gibson TJ, Higgins DG.2007. Clustal W and Clustal X version 2.0. Bioinformatics 23:2947-2948.

Nakao MC, Nakai K. 2002. Improvement of PSORT II protein sorting prediction for mammalian proteins, Genome Inform 13:441-442.

Tang B, Zhang D, Li H, Jiang S, Zhang H, Xuan F, Ge B, Wang Z, Liu Y, Sha Z, Cheng Y, Jiang W, Jiang H, Wang Z, Wang K, Li C, Sun Y, She S, Qiu Q, Wang W, Li X, Li Y, Liu Q, Ren Y. 2020. Chromosome-level genome assembly reveals the unique genome evolution of the swimming crab (Portunus trituberculatus). Gigascience 9:giz161. doi: 10.1093/gigascience/giz161

Teufel F, Almagro Armenteros JJ, Johansen AR, Gislason MH, Pihl SI, Tsirigos KD, Winther O, Brunak S, von Heijne G, Nielsen H. 2022. SignalP 6.0 predicts all five types of signal peptides using protein language models. Nat Biotechnol 40:1023-1025.

Zhu B, Su X, Yu W, Wang F. 2022. What forms, maintains, and changes the boldness of swimming crabs (Portunus trituberculatus)? Animals (Basel) 12:1618.

Round 2

Reviewer 2 Report

Letter to Authors
ijms-2118475-v2
Identification and characterization of 5-HT receptor 1 from Scylla paramamosain: The essential roles of 5-HT and its receptor gene during aggressive behavior in crab species
Xinlian Huang, Yuanyuan Fu, Wei Zhai, Xiaopeng Wang, Yueyue Zhou, Lei Liu, Chunlin Wang

220125

Dear authors,

This revised MS is still not well-written containing many wordy, redundant and verbose phrases. These issues are newly introduced in the revised sentences. You seem trying to show yourself off bigger than real using such verbose phrases. Nevertheless, your revision has somewhat improved so that I could identify points for further improvement. The most important point is unclarity in number of replicates. You state there were 30 replicates in L456, but at least in some experiments, it seem impossible. You must indicate number of individuals tested in each experimental result. The easiest way is to add "n=xx" at each figure picture.
See below for detail.
Citing line numbers are of a pdf file showing track changes.

L59-66
Some researchers have found .. [18]. (simple reference list -> delete

L86
Studies have shown that -> delete

L87
behavior, and the -> behavior. (break sentence here)

L91
[24], ; in addition, -> [24]. (break sentence here)

L97
crustaceans[27,28] -> crustaceans [27,28]

L109
and genus Scylla -> delete

L110
, and it plays a vital role -> delete
the marine -> marine

L112
caused by -> due to

L113
during breeding and production -> delete

L114
mud crabs -> the mud crab
You presented what mud crab at first in L109. You may refer that mud crab as "the mud crab" from later on thoroughout.

L115
mud crabs -> the mud crabs
biological amine -> Ba

L128
5'-UTR -> 5'-untranslated region (UTR)
Spell-out at the first appearance. Abbreviation in L129 is OK.

L131
(pI) (never appear again) -> delete

L139,193,202,220,504,582,592
<i>Sp5-HTR1</i> (product of a gene) -> in Roman

L142,143,146,148(twice),149(twice),150,164
5-HTR1 (gene sequence) -> in Italics
See L155 where deduced AA seqs are deemed as gene seqs.

L155,164,606,607
SP5-HTR1 -> Sp5-HTR1

L161
Figure quality is nothing improved. Fonts are too small to see. Read my comment last time.

L191
conditions (unclear) -> prediction
antibody ? -> antigen ?
clone ? -> product ?

L200 figure 3 picture
Add "n=XX" (number of tissues tested) at each bar in B.

L223 figure 4 picture
Add "n=XX" at each bar in A-D.

L232
mud crab -> the mud crab

L234,240,259
movement speed (unnatural English) -> moving speed

L237,258
movement distance -> moving distance

L245
compared to the control group (redundant) -> delete
See L248.

L246
5-HT (redundant) -> delete
See L248.

L247
, and that pincer force was significantly reduced in the mud crab group injected with (verbose) -> and

L248
5-HT compared to the control group (P < 0.05), -> (P < 0.05) 5-HT compared to the control group, respectively.

L256 figure 5 picture
Add "n=XX" at each bar in B-E.
In E, both NS and 5x10^-7 groups are marked "bc". Is it OK? See L250.

L266
duration of aggression and aggressive intensity (verbose) -> duration and intensity of {aggression, aggressive behavior}

L268
5x10^-7 268 mol/L 5-HT group and 5x10^-5 mol/L 5-HT group (verbose) -> 5x10^-7 mol/L [5-HT] and 5x10^-5 mol/L 5-HT groups
See L271.

L274-275
the aggressive intensity of mud crabs in the control group, saline group, and 5-HT concentration of 5x10^-5 mol/L 5-HT group and 5x10^-3 mol/L 5-HT group
 (verbose) -> that of [all] other experimental and control groups

L279 figure 6 picture
Add "n=XX" (number of pairs tested) at each bar in A,B.

L287,321
SP5-HTR1 -> Sp5-HTR1 (in Roman)

L297
5-HTR1 -> in Italics

L299-301
was cloned by using RACE .. protein was deduced -> was cloned, sequenced, and the molecular weight of the protein was deduced (Fig. 1) (further be compacted)

L305
Multiple amino acid sequence alignment showed that (verbose) -> delete

L319
studied; however, -> studied. However, (break sentence here)

L321
S. paramamosain -> in Roman

L351
is (your result) -> was

L352
5-HT ? -> Sp5-HTR1 (in Roman) ?

L358
cerebral and thoracic ganglion -> cerebral and thoracic ganglia (two ganglia)

L394,401,403
in the -> of the

L396
This may have been related to the concentration of 5-HT injected. (does not make sense) -> delete
See L398.

L398
concentration; a low concentration -> concentration. A low concentration (break sentence here)

L399,436,438
mud crabs -> the mud crabs

L407
that 5-HT contributed -> contribution of 5-HT
Use noun phrases to make it short.

L410-413
the low concentration 5-HT group was .. than those of the control group and high concentration 5-HT group. (redundant) -> the low concentration 5-HT group was {more intense, stronger} and longer than those of the control and high concentration 5-HT groups (Fig. 6).

L424
an increase in 5-HT level increases crustacean aggression ?
This is inconsistent with your results (Fig. 6). Optimum concentration? Pleiotropic effects? See next.

L425
However, ?
When you revise the above, this connecting word is unnecessary.

L440-443
For example, .. the survival rate of individuals. (simple reference list) -> Additives of L-tryptophan, the precursor of 5-HT, to farmed crustaceans increased the content of 5-HT in their hemolymph, inhibited the occurrence of fighting behavior, and improved the survival rate [57-59].

L453
the night -> a night

L456
, with 30 replicates -> , each with 30 individuals ?
The same replicates would be impossible upon selection of males in 4.6.

L457
5 mμg/mLg of 5-HT, 20 μl per crab ?
In this way, you cannot achieve intended doses described in L509.
-> 20 μl of appropriate concentration of 5-HT for doses described later ?

L460
cerebral -> cerebellar

L472
Invitrogen -> Invitrogen, Waltham, MA, USA

L502
KLH and BSA -> keyhole limpet hemocyanin (KLH) and bovine serum albumin (BSA)
You may omit spelling-out of BSA, because it is so commonly used.

L508
S. paramamosain -> in Roman

L527
[11,74] -> delete

L558-593
These two sub-sections should move before 4.5 to fit with the order in the result section.

L558
Sp5-HTR1 -> in Roman
S. paramamosain -> in Roman

L559
Individual crabs ?
30? See L456.
21 (3x7)? See L562.

L575-577
Statistical analysis .. statistically significant. (redundant) -> delete

L578
4.8. Western blotting -> no indent, in Italics

L602
5-HTR -> in Italics

L613
10^-3 -> 10<sup>-3</sup>
10^-5 -> 10<sup>-5</sup>
10^-7 -> 10<sup>-7</sup>
You seem unable to understand basic power signs. I hope you understand html signs.

L626
the National Center for Biotechnology Information (NCBI) Sequence Read Archive ?
Your sequence accession seems of a NCBI GenBank database. You did not performed massive parallel sequencing, which data would be stored in a read archive.

L638 references
Check the reference list carefully again from the beginning. Reference lists are frequently hotbeds of errors. You might add, omit or swap citation in the main text on the way internal revision. Numbering of the references might then shift. If so, readers think you are making irrelevant citation. It is the authors' responsibility that all references are properly cited.

L639,etc (many)
Make sure if journal title words are abbreviated when possible. Check thoroughly.
Zoological Research -> Zool. Res.

L652
sp. -> in Roman

L667,etc
Make sure if abbreviated journal titles are in title case. Check thoroughly.
behavior -> Bahav.

L708
genomics -> Genom.

Round 3

Reviewer 2 Report

Letter to Authors
ijms-2118475-v3
Identification and characterization of 5-HT receptor 1 from Scylla paramamosain: The essential roles of 5-HT and its receptor gene during aggressive behavior in crab species
Xinlian Huang, Yuanyuan Fu, Wei Zhai, Xiaopeng Wang, Yueyue Zhou, Lei Liu, Chunlin Wang

230205

Dear authors,
You have well revised your MS. Only very minor points left for improvement. See below for detail. Citing line numbers are of a pdf file showing change history.

L202,226,282
, n = 3 (not a complete sentence) -> (n = 3)

L258
, n = 9 (not a complete sentence) -> (n = 9)

L444,447
Delete either of redundant "[55,70,71]".

Author Response

Responses to Reviewers

To Reviewer 2

Question: Page 10; line 202 “The bars (B) correspond to the standard deviation (SD) of the mean values, n = 3.”

Response: Thanks. This sentence has been revised to “The bars (B) correspond to the standard deviation (SD) of the mean values (n = 3).”

Question: Page 11; line 225 “The bars (A-D) correspond to the standard deviation (SD) of the mean values, n = 3.”

Response: Thanks. This sentence has been revised to “The bars (A-D) correspond to the standard deviation (SD) of the mean values (n = 3).”

Question: Page 12; line 257 “The bars (B-E) correspond to the standard deviation (SD) of the mean values, n = 9.”

Response: Thanks. This sentence has been revised to “The bars (B-E) correspond to the standard deviation (SD) of the mean values (n = 9).”

Question: Page 13; line 281 “The bars (A) correspond to the standard deviation (SD) of the mean values, n = 3.”

Response: Thanks. This sentence has been revised to “The bars (A) correspond to the standard deviation (SD) of the mean values (n = 3).”

Question: Page 16; line 444,447. Delete either of redundant "[55,70,71]".

Response: Thanks. This sentence has been revised to “Additives of L-tryptophan, the precursor of 5-HT, to farmed crustaceans increased the content of 5-HT in their hemolymph, inhibited the occurrence of fighting behavior, and improved the survival rate [55,70,71].”
